# Therapist perceptions of the Danish Physiotherapy Research Database for assessing patients with chronic disease

Peter W. Stubbs[1], Henriette H. Stabel[2], Nils-Bo de Vos Andersen[3], Helle Rønn Smith[4], Erhard T. Næss-Schmidt[2]*

1 Graduate School of Health, Discipline of Physiotherapy, University of Technology Sydney, Sydney, Australia, 2 Hammel Neurorehabilitation Centre and University Research Clinic, Aarhus University, Aarhus, Denmark, 3 Primary Health Care and Quality Improvement, Viborg, Denmark, 4 VIA University College, Holstebro, Denmark

* erhnae@rm.dk

**Data Availability Statement:** All relevant data are within the manuscript and its Supporting Information files.

## Abstract

### Background

The Danish Physiotherapy Research Database for chronic patients receiving Free of Charge Physiotherapy (PhysDB-FCP) was piloted over a 1-year period. The purpose of the PhysDB-FCP is to provide a user friendly digital online structured tool that standardizes initial and follow up clinical assessments generating data that can be used for clinical decision making and support future research in physiotherapy for patients with chronic disease. Although initial assessments were completed, the attrition rate was 73% and 90% at 3- and 6- months, respectively, which suggests problems with the current tool.

### Objective

To evaluate the perspectives of the physiotherapists that used the PhysDB-FCP and propose changes to the tool based on this feedback.

### Materials and methods

Fifty of the 103 physiotherapists introduced to the PhysDB-FCP completed an anonymous online survey. Physiotherapists were asked Likert/categorical and yes/no questions on experiences with the PhysDB-FCP within their practice, perceptions of patient experiences, suitability of the resources and support provided by the PhysDB-FCP working group and the ideal administration frequency of the assessments within the PhysDB-FCP. Open ended feedback on possible improvements to the PhysDB-FCP was also collected.

### Results

Physiotherapists agreed that the PhysDB-FCP was useful for taking a physiotherapy assessment (74%) and the patient survey was useful for goal setting (72%). Although physiotherapists felt the PhysDB-FCP was well-defined (82%), only 36% would like to use a similar tool again. Generally, the PhysDB-FCP was too time-consuming, administered too

**Funding:** EN was supported by the Foundation of research, quality and education in practical physiotherapy, Denmark [1334]. The initial setup of the PhysDB-FCP was supported by the Practice Physiotherapy Quality Fund. The funders had no role in study design, data collection and analysis, decision to publish, or preparation of the manuscript.

**Competing interests:** The authors have declared that no competing interests exist.

frequently and included irrelevant items. For example, 72% of physiotherapists took >45 min to administer the assessment in the first consultation which was performed over multiple sessions.

## Conclusions

The perspectives of physiotherapists using The PhysDB-FCP suggest specific changes that will ensure better use of the tool in future practice. Changes will likely involve administering the assessment less frequently (every 6-months to 1-year), shortening the assessment, and using diagnosis-specific assessment items.

## Introduction

Worldwide, people are living longer and the number of people living with chronic conditions (and health issues) are increasing [1, 2]. As such, there is a substantial future challenge to ensure that people are able to live as functionally as possible, with the health issues they encounter, at an appropriate (private / public / societal / financial) cost. Although, management of chronic conditions is becoming more effective, it is often costly, with a greater requirement to document and prioritize the most effective and least costly (treatment) approaches. Recently, national and international clinical guidelines for both assessment and interventions for people with chronic conditions have emerged [3–6], and there is a comprehensive evidence that exercise is effective for increasing and maintaining function in people with chronic conditions [7]. Given this, health practitioners advocating for exercise are critically important in the management of patients with chronic disease.

Systematic monitoring of different interventions in primary care could be helpful as such data would allow for population-wide decisions based on function (and other) outcomes. For this to occur, infrastructure and systems to enable easy data entry, summary/analysis, access and synthesis are required. Numerous databases and registries have been created in a number of countries for a number of conditions [8–13]. However, to date, systematic collection/dissemination of data on physical (including activities of daily living and participation), social and psychological function in patients receiving long-term physiotherapy in outpatient clinics is sparse, although not completely absent [11, 13]. For any database to be effective, its content needs to be relevant for clinical users, stakeholders and citizens/patients to ensure (near to) complete and accurate data.

To systematically monitor patients receiving Free of Charge Physiotherapy (FCP) [13], we designed and piloted the Danish Physiotherapy Research Database and data collection platform (PhysDB). Prior to piloting, physiotherapists using FCP have reported that they would like guidelines to assist in the reporting of assessment data to the General Practitioner (GP), as it is currently difficult to ascertain exactly what information is required for patients receiving FCP [14]. As such, the purpose of the PhysDB-FCP was to 1) create a clinically relevant tool that could provide information about function and disability from a biopsychosocial perspective and 2) provide data to facilitate population-wide decisions on the cohort receiving PhysDB-FCP to make decisions about the program. To achieve this, a standardized assessment and evaluation tool of patients receiving FCP has been created to facilitate the easy imputation, extraction and synthesis of clinical data which will facilitate research and enable a continuous description of the FCP population. The hope is to improve the quality of Physiotherapy and create room for (joint) reflexivity on rehabilitation. A similar database has recently been

developed/tested in Norway, where patients receiving primary healthcare have physiotherapy management and are also assessed using patient-reported questionnaires on disability, pain and psychosocial factors [11]. Other databases have also been developed but often include different patient populations, that are not in primary care and do not include data from patient reported questionnaires combined with extensive physical assessments [8–10, 12].

Eleven clinics participated in piloting the PhysDB-FCP, conducted over a one-year period. Of the 534 patients that performed baseline assessment, 142 patients were assessed at 3 months and 52 were assessed at 6 months [13]. Although participation was voluntary, and assessors were instructed to perform an assessment at 3 and 6 months, the attrition rate was unexpectedly high. In the development of the PhysDB-FCP highly dedicated and experienced clinicians were consulted however these clinicians may have been different to the clinicians using the tool. It may also demonstrate the gap between clinical users and stakeholders in terms of the future use, necessity and prioritising of resources for the PhysDB-FCP [13]. For the PhysDB-FCP to be effective, and for further implementation regionally or nationally, we need to understand the problems associated with the PhysDB-FCP and make the necessary adjustments to ensure clinicians understand the future use, necessity and possibilities associated with the PhysDB-FCP and want to 'buy in' to the tool. Although suggestions were proposed in Næss-Schmidt et al. [13], these required further investigation through consultation with the physiotherapists that used the PhysDB-FCP tool.

As such, the purpose of this study was 1) to evaluate the perspectives of the physiotherapists that used the PhysDB-FCP during the one-year piloting period and 2) preliminarily propose changes to the tool based on this feedback which will be discussed further with stakeholders and clinical users.

## Materials and methods

### Context

A detailed description of FCP and the PhysDB-FCP is provided in Næss-Schmidt et al. [13]. FCP is provided by physiotherapists to patients with chronic or progressive disease, with a range of musculoskeletal and neurological diagnoses. Primary diagnoses include Multiple Sclerosis (22.7%), Parkinson's disease (17%), Stroke (9.6%) and Chronic Arthritis (8.6%). Patients are entitled to a limited number of sessions (40 sessions per year is the upper limit) which includes one-on-one or group therapy sessions, depending on their specific requirements and the severity of their condition. The Danish Ministry of Health registers a limited number of physiotherapists to provide FCP. A condition of registration with FCP is that physiotherapists must provide a report on each patients functional status to the referring general practitioner once per year or if substantial disease changes occur. The tool was created iteratively with input from a workgroup, which included both clinical and research stakeholders. Physiotherapy clinics in the Central Region of Denmark volunteered to partake in the piloting of the PhysDB-FCP. Each clinic received training on how to use the PhysDB-FCP, as the software was a novel standalone platform. The PhysDB-FCP consisted of a physiotherapy assessment including demographic information (name, age, sex, assessment date, year of diagnosis), health status (pain (location and numeric rating), medication, height, mass, involuntary weight loss and speaking/swallowing problems), questions on daily functioning (information about personal and instrumental activities of daily living, use of and type of assistive device, dizziness and fatigue), obligatory functional tests (Timed up and go test, Sit to stand test, walking test (either 6 minute walk test, 10 meter walk test or 40 meter walk test), the box and block test (if indicated)) and documentation of a treatment plan (goal setting, type of care received (individual, team-based, combined, supervised/unsupervised), expected time(s)/effect(s) of treatment,

patient approval of treatment plan and reporting of plan to General Practitioner) [13]. A patient survey, usually administered prior to the physiotherapy session, consisted of individual questions (on civil status, education, work status, amount of sick leave, fear of falling and sleep quality) and validated questionnaires (EQ- 5D- 5L, WHODAS 2.0–12 item, WHO-5) [13].

## Participants

A total of 11 private outpatient clinics participated in piloting the PhysDB-FCP during August 2017 to January 2019 [13]. All physiotherapists in the clinics (n = 103) were invited to participate in an anonymous survey via email. The email provided a link to a survey evaluating the PhysDB-FCP. After the original email, all physiotherapists were sent a reminder email at 2 weeks and 4 weeks.

## Survey of clinicians using PhysDB-FCP

To determine the perceptions of clinicians using the PhysDB-FCP tool, we conducted a survey to assess the implementation, relevance and applicability/usefulness of the pilot program. As the survey was developed to determine areas of improvement or further development for possible future implementation of the PhysDB-FCP, we utilised and modified the framework from Huijig et al. [15], using questions for areas we required feedback on. The survey was translated to Danish (and questions delivered / responses were provided in Danish) but for the purpose of this manuscript, responses were translated to English with two bilingual speakers confirming the translation. The initial survey was trialled by three Danish physiotherapists and the survey was altered based on their feedback.

We collected general information on physiotherapists that performed the assessment (Table 1). For evaluating the PhysDB-FCP, Likert scale, yes/no and categorical questions were asked around the areas of attitudes towards the: physiotherapists experience with the PhysDB-FCP, physiotherapist perceptions of patient experiences, resources and support, ideal

Table 1. Characteristics of the physiotherapists that completed the survey (n = 50) about using the PhysDB-FCP in patients with receiving FCP.

| Characteristics | |
|---|---|
| Age (yr), mean (SD) | 43.9 (10.7) |
| Missing (%) | 2 |
| Gender, female (%) | 54 |
| Time as PT (yr), mean (SD) | 15.2 (9.6) |
| Role (%) | |
| Owner | 24 |
| Manager | 4 |
| Employee | 14 |
| Self-employed | 56 |
| Other | 2 |
| FCP patients treated over the pilot period (%) | |
| 1–5 | 50 |
| 6–10 | 22 |
| 11–15 | 14 |
| 16–20 | 10 |
| >21 | 4 |

FCP = Free of Charge Physiotherapy; PT = Physiotherapist.

frequency of PhysDB-FCP use. An English version of the questionnaire is provided in the supplementary material (S1 Table).

Likert scale responses ranged from 1–5 corresponding to answers of strongly agree (1), agree (2), neither agree or disagree (3), disagree (4), strongly disagree (5). In addition, we also asked an open-ended question: *"Do you have any suggestions for changes or have you experienced anything that could make PhysDB-FCP a better assessment and evaluation tool?"*

## Data analysis and statistics

Clinician characteristics were reported using descriptive statistics. The percentage responses for all survey questions were presented. Suggestions for improvements from answers to the open-ended questions were assessed and ideas were categorized. The frequency that each idea was suggested was tallied and reported. Missing data has been highlighted when it occurred.

## Ethical approval

The study was conducted in accordance with the declaration of Helsinki and all participants provided written informed consent. The study was approved by the Danish Data Protection Agency (j.nr. 1-16-02-757-17).

## Results

Of the 103 clinicians that were surveyed, 63 clicked on the link, 58 completed the survey and 55 provided consent to use their responses. An additional five participants were identified as having not used the PhysDB-FCP tool and their responses were removed. The responses of 50 respondents were included in the study. For the quantitative questions, each question had a maximum of 4% missing data. Twenty respondents answered the open-ended question.

## Demographic information

Table 1 shows the characteristics of the physiotherapists that completed the survey (n = 50). All participants had a physiotherapy qualification and four participants had an additional diploma. Twenty-six percent stated they used other functional tests not included in the PhysDB-FCP however 16% did not state which tests were used and 4% stated tests that were already included in PhysDB-FCP (Timed Up and Go and Berg balance scale). Six percent used other tests not used in the PhysDB-FCP.

## Physiotherapists experience with the PhysDB-FCP

Fig 1 shows the levels of agreement in statements about the purpose and usefulness of the PhysDB-FCP. Eighty-two percent of therapists agreed that the purpose of the PhysDB-FCP was well-defined, 78% agreed that the patient survey was useful and 74% agreed that the PhysDB-FCP was useful for a baseline assessment. Sixty percent agreed that it was useful to document change in function and 56% found the automatic summary useful. Forty-four percent of respondents stated they would not like to use a tool similar to the PhysDB-FCP in the future, 16% stated that they might, 36% stated they would and 4% didn't know.

Fifty-four percent of physiotherapists took 45–60 min to perform the baseline assessments, with 16% and 2% of physiotherapists taking 60–75 min and >75 min, respectively. Baseline assessments were frequently split over multiple consultations, which occurred often or always for 44% of physiotherapists, sometimes for 20% of physiotherapists and rarely or never for 36% of physiotherapists. The functional follow up assessments took <30 min for 86% of

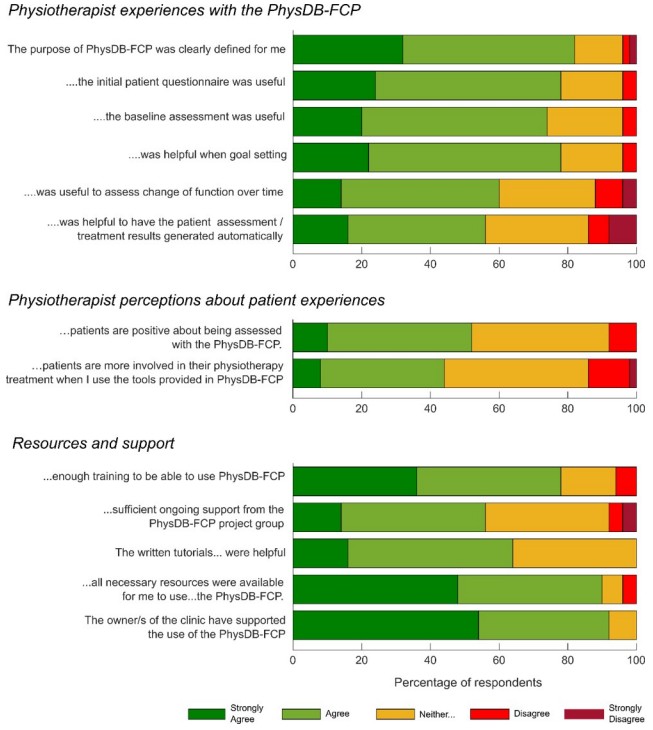

**Fig 1. Therapist responses for all Likert scale questions.** Included are responses that dealt with the Physiotherapist experiences with the PhysDB-FCP-FCP, responses that dealt with the physiotherapist perceptions of patient experiences when using the PhysDB-FCP-FCP and responses that dealt with resources and support. Reponses categories were: strongly agree (dark green), agree (light green), neither agree nor disagree (yellow), disagree (red), strongly disagree (dark red).

physiotherapists and 30–45 min for 12% of physiotherapists (Note: There was missing data for 1 physiotherapist).

## Physiotherapist perceptions about patient experiences

Fig 1 shows the level of agreement with statements about physiotherapist perceptions about patient experiences. Fifty-two percent of physiotherapists agreed that patients were positive about being assessed by the PhysDB-FCP and 44% agreed that patients are more involved in their physiotherapy treatment due to the PhysDB-FCP.

## Resources and support

Ninety-two percent of respondents were taught to use the PhysDB-FCP tool by the PhysDB-FCP team, 4% were taught by colleagues and 4% were not taught to use the tool. Fig 1 shows the level of agreement with statements about resources and support. For 78% of respondents the PhysDB-FCP team provided enough training initially, although only 56% agreed that there was enough ongoing support and 64% agreed that the written tutorials were useful. The majority agreed that the practice had the necessary resources to use the PhysDB-FCP (90%) and that the owners of the practice were supportive of the PhysDB-FCP (92%).

## Ideal frequency of PhysDB-FCP use

For the comprehensive (baseline) assessment, physiotherapists felt that the ideal frequency for using a tool similar to the PhsyDB-FCP was every 12 months (76%), every 6 months (14%) or

other—unspecified (10%). For the shorter (follow up) assessments, physiotherapists felt that the ideal frequency for using a tool similar to the PhsDB-FCP was every 3 months (14%), every 6 months (58%), every 12 months (22%) or every 12 months—but patient dependent (2%).

## Open-ended question

Twenty respondents answered the open-ended question. Table 2 shows the suggestions for improvement to the PhysDB-FCP by the respondents, the number of respondents that suggested the improvement and example quotes for each suggestion. The main suggestions were that the assessment 1) was too long/comprehensive and time-consuming and 2) required integration with current clinical systems.

## Discussion

This study assessed experiences of physiotherapists using the PhysDB-FCP platform for the assessment of patients receiving FCP. Although responses were generally positive, when asked if they would like to use a similar tool in the future, responses were mixed with 44% of physiotherapists stating they would not want to use the tool, 20% uncertain and 36% stating they would. This indicates that the tool requires improvement and discussion with frontline clinicians who used the PhysDB-FCP. The questionnaire provides potential reasons for the high attrition at 3- and 6-months and provides the basis for further feedback, in the form of interviews addressing database content, barriers/facilitators and purpose.

The physiotherapists that answered the survey were generally middle aged, well experienced in their role/ employment. The size of the clinics represented the common distribution of outpatient clinics in The Middle Region of Denmark [16]. These results are likely generalizable to

**Table 2. Suggestions for improvement to the PhysDB-FCP by the respondents and example quotes for each suggestion.**

| Respondent suggestions for improvement | Example text |
|---|---|
| Physiotherapist experience with the PhysDB-FCP<br>• The assessment was long / comprehensive and time consuming to administer (7), repetitive (1) and complex (1)<br>• The assessment was non-specific to the types of patients being tested (2)<br>• Requirement for integration with current systems/ tools (6)<br>• Disliked the automatically generated patient summary (4)<br>• Didn't receive patient retest reminders (2)<br>• Didn't receive notifications that patients hadn't completed the patient survey (1)<br>• Disliked the structure of the assessment form (2) | • *"I think the assessment is time consuming, it must be shorter to be feasible in daily practice, especially for the progressive [less-severe] group"*<br>• *"The assessment needs integration with the journal systems already used in the clinic, so the data is automatically transferred and can be opened within normal data system of the clinic"*<br>• *"Needs a better conclusion / resume of the results for use in an evaluation of the patient"*<br>• *"The re-test reminders do not work."*<br>• *"A reminder to the clinicians if the patient doesn't answer the surveys"*<br>• *"There is a wish to make a more schematic structure"* |
| Physiotherapists perception of the patients experience<br>• No added value compared to current clinical assessments (1)<br>• Negative patient perceptions (1) | • *"I have only assessed already existing patients at the clinic and nobody found any meaning in being assessed with it."* |
| Resources and support<br>• Insufficient training (1)<br>• Insufficient communication from the PhysDB-FCP team on ongoing additions to the PhysDB-FCP (1) | • *"Not enough experience with the assessment"*<br>• *"A need for continuous information/ communication about changes in the database so the clinicians are prepared before they make a new assessment."* |
| Ideal frequency of PhysDB-FCP use<br>• Testing was too frequent (1) | • *"...The tests are too close to each other..."* |

The number of respondents that made the suggestion are denoted in brackets.

other outpatient clinics with experienced physiotherapists. However, 50% of physiotherapists only assessed 1–5 patients during the pilot phase. Therefore, results of this study may be influenced by lack of experience with the PhysDB-FCP tool, among other challenges discussed in the following sections.

## Potential changes required

**Physiotherapists experience with the PhysDB-FCP.** General feedback was that the tool was too time-consuming to administer and too comprehensive with 9/20 comments specifically addressing this issue. This was corroborated by the information that the assessment was frequently split over multiple days with 96% of physiotherapist having done this with at least one patient. Only 28% of physiotherapists could perform the comprehensive baseline assessment in < 45 min, with 72% taking > 45 min, which may in part be a result of poor training or lack of experience using the assessment tool and database. As physiotherapists receive government-set compensation for the initial consultation for patients undergoing FCP (covering ≈45 min), having an assessment that is too time consuming may reduce compliance and enthusiasm for the tool. Although the assessment is time-consuming, the tool develops an automatic generated summary which offset some of the time lost. Despite this, no physiotherapists commented on this aspect.

This combined with the two comments regarding the lack of assessment specificity and irrelevance of some items for some patients may be the reason for the mixed responses in wanting to use the PhysDB-FCP in the future. This could also be the reason that only 60% of physiotherapists believed it was useful to document change in function. Although the intention was to standardize patient assessments and provide systematically collected and easily summarized information to both clinicians, researchers and policy makers, it may have resulted in information that was too generic and inappropriate for some patients. Moving forward, we will need to shorten the generic section and prioritize questions and clinical measures based on clinical usefulness for individual patient severities / diagnoses and stakeholder (patient, clinician, policy maker) interests. Additional assessment options tailored to patient severity and diagnoses may also be important. For some patient populations there are already core outcomes sets and for other outcome sets are being developed [3–6]. This adaption will not only increase the relevance of the tool for the patients but it would also facilitate comparisons between specific patient groups, internationally.

The open-ended questions demonstrated that the lack of integration with current systems (already in place within the clinic) was problematic. For the physiotherapists, the lack of integration with current systems resulted in double entry of data i.e. one entry for the clinic system and another entry for the PhysDB-FCP in terms of planning and assessment. Further, only 56% agreed that the automatic generation of results was useful. The automatic generation of results needed to be opened and copy-pasted into the existing systems within the clinic, which is time-consuming. Moving forward, either the PhysDB-FCP will need to be integrated into current clinical systems and be able to extract data from these or the PhysDB-FCP is to be used as the sole system within the clinic. Both options would be problematic and would require significant changes for either the database or clinical platform. Physiotherapists will need to be further consulted to assess which options are most desirable, and how we could overcome this issue in the future.

**Physiotherapist perceptions of patient experience.** In general, physiotherapists perceived that many patients were not more positive (52% were more positive) or more involved (44% were more involved) when compared to standard tools used in the clinic. This does not necessarily mean that the physiotherapists perceived that patients disliked the tool, but more

that it was perceived as not more beneficial than current assessment procedures. Only one open-ended answer addressed the issue directly: *"I have only assessed already existing patients at the clinic and nobody found any meaning in being assessed with it."* If physiotherapists and patients have an existing clinical relationship, a number of the items are unnecessary to measure every 3-months, especially if a clinical picture has already been established. If patients and physiotherapists do not see any meaning with such a comprehensive assessment, then it is difficult to justify using the PhysDB-FCP, at the frequency performed during piloting. As such, a better justification of the aims and purpose of the PhysDB-FCP may be required to increase 'buy in'. As also indicated in other aspects of the survey, it may be necessary to reduce the generic part of the PhysDB-FCP to a few important items and make the tool more flexible so that the physiotherapist can individualize it to a patient needs. To elaborate on the patient perspective, we plan to interview a group of different patients who experienced the PhysDB-FCP in a future study.

**Resources and support.** In general, physiotherapists felt they had been provided enough training (78%) however only 56% felt they had been provided sufficient ongoing support and 64% found the written tutorials useful. As such, ongoing communication/support requires improvement. During piloting, physiotherapists were provided one 2–3 hour introduction to PhysDB-FCP and were encouraged to contact the project lead by email or phone if any problems occurred. This service was only used a limited number of times by physiotherapists, with the project lead receiving 1–2 contacts per week. Thus, further evaluation and in-depth feedback is required on the tutorials delivered and support provided.

**Ideal frequency of PhysDB-FCP use.** The majority of physiotherapists reported that the assessment was administered too frequently. During piloting of the PhysDB-FCP, we asked physiotherapists to administer the tool at baseline (comprehensive baseline assessment), 3 months (shorter follow up assessment) and 6 months (shorter follow up assessment). When asked about the ideal frequency of assessments the majority (76%) of respondents felt that the comprehensive baseline assessments should be administered every 12 months. For the follow up assessment, the majority of participants (82%) felt it should be administered every 6 months or every year. The frequent assessments may have resulted in the non-prioritization at the 3- and 6- month assessment timepoints. In the future, we will need to consider the purpose of the follow-up assessment. If the assessment should be used for characterizing the cohort in general, reducing the frequency of administration to every 12 months may be beneficial. However, for individual and interventional evaluation it may be better to be administered after a specific event. Therefore, a standard 12 month assessment with flexible self-chosen assessments (in between) may be the best solution to address clinical relevance, wider decision making and cohort descriptions.

## Current strengths

There were some strengths identified from the current questionnaire. Eighty-two percent of respondents understood the purpose of the PhysDB-FCP. Seventy-eight percent found that the patient survey (usually administered before the assessment) was useful. This is important as comprehensive patient surveys are rarely performed in databases and also provide valuable information, not usually captured by databases, for clinicians, researchers and decision makers. Furthermore, 78% of therapists found the tool useful for patient goal setting which is an area that has previously been reported as difficult in Danish rehabilitation settings for patients receiving FCP [14] and internationally [17]. Furthermore, the patient survey can provide important information, sometimes not routinely asked, prior to a consultation to enable the formulation of better therapist-patient developed patient-centered goals. The majority of the

practitioners had the necessary knowledge and physical resources to perform the assessment (90%) and had supportive owners (92%). For 'buy in' to any tool and sustainability of new initiatives, support of owners is imperative [18]. Although these statements are encouraging and indicate that further implementation of the tool on a larger scale is possible, as 44% would not like to use the tool in the future, further feedback and changes to the tool are required before implementation into clinical practice.

## Limitations

Only ≈50% of the total physiotherapists approached answered the survey and we do not know why these surveys were not completed. As such, responses reflect a self-selected group that may be more positive or negative than all physiotherapists using the tool. Although the evaluation is specific to the PhysDB-FCP, the findings are likely generalizable to the development of other similar databases and these results will likely assist other people attempting similar projects. The survey used was adapted from the framework of Huijig et al. [15] and there could be other interesting perspectives not captured by our survey. However, we asked questions that we specifically required feedback on, but acknowledge the survey could have been more comprehensive. Despite this, an open-ended question regarding suggestions for changes or things to make the assessment and evaluation tool better was asked.

Not all physiotherapists were positive about using the PhysDB-FCP, and through our survey, we may have only captured those most engaged with the tool. Given this, there is the possibility that less motivated/engaged physiotherapists may have imputed/coded data erroneously. This would affect data quality within the database and will need to be acknowledged/considered in future studies using the database. With any database, there are possible issues with data quality, and this will contribute to extraneous data noise. However, in our study, participation was entirely voluntary, and if physiotherapists didn't want to use the tool, they weren't required to. As such, it is likely that less motivated/engaged clinicians, would have not filled in specific items or decided not to use the tool. Further, data for each patient appeared internally consistent (i.e. patients with slow timed up and go times had fewer sit to stand repetitions in 30 seconds) increasing our confidence in the data. Although the current study identified some issues, we are glad we have identified these. Having this knowledge is valuable in identifying potential sources for change that will likely increase future compliance and data quality. Changes such as a) integrating the PhysDB-FCP into current systems so that only one data input is required, b) having a shorter more basic tool allowing physiotherapists to complete the tool in 45 min (the time physiotherapists are compensated for), and c) increasing resources for education and ongoing support, will inevitably improve compliance, subsequent data quality and reduce missing data.

## Future directions

The current survey has provided some information on physiotherapist perceptions of the PhysDB-FCP tool. It preliminarily identifies numerus issues with the PhysDB-FCP tool, its administration and ongoing support that will be explored further before the tool is implemented further. To do this, we are performing focus groups with physiotherapists from different clinics and securing a patient perspective through individual patient interviews. The results of the survey presented in the current study will be used as the framework for questions during these focus groups and interviews, and will guide modifications to PhysDB-FCP. For example, although 82% of respondents understood the purpose of the PhysDB-FCP, 44% stated that they would not want to use a tool similar to this in the future. This means respondents may not have agreed with or liked the purpose. We will need to ascertain why physiotherapists did not want to use a similar tool in the future. Furthermore, as physiotherapist

perceptions of patient experiences indicated that some patients did not feel positive or more involved with the PhysDB-FCP compared to standard assessments, it is imperative to gather patient perspectives. Providing a patient perspective will also allow us to identify potential issues with how the tool is used for shared (therapist-patient) decision making and creation of patient-centred goals. Once patient and physiotherapist responses have been gathered and synthesized, we will need to present the issues and suggested changes with other stakeholders (decision / policy makers) and ensure that any changes made based on physiotherapist and patient feedback aligns with the agreed upon purpose of the PhysDB-FCP platform. This purpose is to 1) create a clinically relevant tool that can provide information about functioning and disability from a biopsychosocial perspective and 2) provide data to facilitate population-wide decisions on the cohort receiving PhysDB-FCP to make decisions about the program. Finally, a re-pilot of the tool will be necessary to assess if the correct changes have been implemented and issues have been resolved from a clinician and patient perspective while satisfying the needs of other stakeholders.

The importance of focusing on implementation setting and considering the interests of different stakeholders, meaningfulness of an intervention/tool and the process of implementation (planning evaluation and reflection) is important [18]. The current study and previous studies [11, 13] show the challenges of implementing an assessment tool in primary care in a heterogeneous patient cohort. Similarly to the current study, in a previous study on a different database [11], therapists did not always use the proposed tool with only ≈50% of eligible patients being assessed with the tool. In that study [11], similar to our previous work [13], reasons for non-compliance were proposed but not formally investigated. The current study explains why implementing a database is challenging and provides reasons and potential issues that can be considered when implementing similar tools in the future.

## Conclusions

The current study has provided areas of improvement for the PhysDB-FCP tool and provided potential reasons for the high testing attrition rate at 3- and 6- months during piloting of the tool. This knowledge could be useful for the development of similar databases. The study highlights that the PhysDB-FCP requires further development with input from all stakeholders (patients, frontline physiotherapists, policy makers and researchers) to ensure the tool is useable in practice. Changes will likely involve administering the assessment less frequently (every 6 months to 1 year), shortening the assessment, integrating the assessment with current platforms and using diagnosis-specific assessment items. We will seek further feedback in the form of interviews addressing the database content, barriers/facilitators and purpose for further iteration of the tool.

## Supporting information

**S1 Table. Survey questions.** The English version of the survey questions in relation to determining physiotherapists experiences when using the PhysDB-FCP. Some domains are modified on the basis of the framework by Huijg et al. [15].
(DOCX)

**S1 File. Dataset for the study.**
(XLSX)

## Acknowledgments

We would like to thank the physiotherapists who were part of this study.

## Author Contributions

**Conceptualization:** Peter W. Stubbs, Nils-Bo de Vos Andersen, Erhard T. Næss-Schmidt.

**Data curation:** Peter W. Stubbs, Henriette H. Stabel, Nils-Bo de Vos Andersen, Helle Rønn Smith, Erhard T. Næss-Schmidt.

**Formal analysis:** Peter W. Stubbs, Erhard T. Næss-Schmidt.

**Funding acquisition:** Erhard T. Næss-Schmidt.

**Investigation:** Erhard T. Næss-Schmidt.

**Methodology:** Peter W. Stubbs, Henriette H. Stabel, Nils-Bo de Vos Andersen, Helle Rønn Smith, Erhard T. Næss-Schmidt.

**Project administration:** Nils-Bo de Vos Andersen, Erhard T. Næss-Schmidt.

**Resources:** Erhard T. Næss-Schmidt.

**Software:** Peter W. Stubbs, Erhard T. Næss-Schmidt.

**Supervision:** Henriette H. Stabel, Helle Rønn Smith, Erhard T. Næss-Schmidt.

**Validation:** Peter W. Stubbs, Henriette H. Stabel, Nils-Bo de Vos Andersen, Helle Rønn Smith, Erhard T. Næss-Schmidt.

**Visualization:** Peter W. Stubbs, Nils-Bo de Vos Andersen.

**Writing – original draft:** Peter W. Stubbs, Erhard T. Næss-Schmidt.

**Writing – review & editing:** Peter W. Stubbs, Henriette H. Stabel, Nils-Bo de Vos Andersen, Helle Rønn Smith, Erhard T. Næss-Schmidt.

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
