## [Decision Letter · Decision Letter 0]

15 Jul 2021

PONE-D-21-09985

Therapist perceptions of the Danish Physiotherapy Research Database for assessing patients with chronic diseases

PLOS ONE

Dear Dr. Næss-Schmidt,

Thank you for submitting your manuscript to PLOS ONE. After careful consideration, we feel that it has merit but does not fully meet PLOS ONE’s publication criteria as it currently stands. Therefore, we invite you to submit a revised version of the manuscript that addresses the points raised during the review process.

We look forward to receiving your revised manuscript.

Kind regards,

Sinan Kardeş, M.D.

Academic Editor

PLOS ONE

Journal Requirements:

Reviewers' comments:

Reviewer's Responses to Questions

**Comments to the Author**

1. Is the manuscript technically sound, and do the data support the conclusions?

Reviewer #1: Partly

Reviewer #2: Yes

2. Has the statistical analysis been performed appropriately and rigorously? 

Reviewer #1: Yes

Reviewer #2: I Don't Know

3. Have the authors made all data underlying the findings in their manuscript fully available?

Reviewer #1: Yes

Reviewer #2: Yes

4. Is the manuscript presented in an intelligible fashion and written in standard English?

Reviewer #1: Yes

Reviewer #2: Yes

5. Review Comments to the Author

Reviewer #1: Thank you for inviting me to review this article. I must commend the authors for evaluating the PhysDB-FCB tool via physiotherapist (PTs) perspectives. This tool/database promises to provide systematised information on chronic patients for clinicians and policymakers. I have few comments that authors could address

Abstract

Authors should state in the background of the abstract what PhysDB-FCB is used for.

The statements from 35-38 were difficult to read. Does this mean that PTs responded to questions on attitude towards 1) experiences with the PhysDB-FCP, 2) perceptions of patient experiences, 3) resources and support and 4) ideal frequency of use- using Likert scale, and yes or no. They also had a comment section to suggest areas requiring improvement on the scale. Can the authors revise?

General.

Authors should provide similar databases across the globe that could be compared with PhysBD-FCB? If there is none, the authors should state it.

Even though the PhysDB-FCB description has been published in detail elsewhere, a summary description of the tool/database will be great. For instance, examples of the obligatory functional test (e,g outcome measures tools, SPPB etc.).

What is the global implication of the findings in this study. The authors stated this, but they did a similar tool/database to compare?

The figures were difficult to read.

Reviewer #2: If Physiotherapists are not very confident in the use of this database, there is a huge possibility of erroneous data input, coding, and misinterpretation of outputs. This paper could be strengthened if authors discuss the implication of their findings on the results of past, current research studies based on this database.

6. PLOS authors have the option to publish the peer review history of their article (what does this mean?). If published, this will include your full peer review and any attached files.

Reviewer #1: No

Reviewer #2: No

---

## [Author Response · Author response to Decision Letter 0]

24 Aug 2021

Reply to reviewers

Based on the comments of both reviewer 1 and 2, we have made amendments to the manuscript. We would like to thank both reviewers for their suggestions.

In the tracked changes document, yellow denotes insertions from the original submission and turquoise with a strike through indicates deletions from the original submission. 

Reviewer: 1

I must commend the authors for evaluating the PhysDB-FCB tool via physiotherapist (PTs) perspectives. This tool/database promises to provide systematised information on chronic patients for clinicians and policymakers. I have few comments that authors could address

Thank you for your suggestions. We have attempted to address all of your comments. We hope the manuscript reads better as a result of this. 

Authors should state in the background of the abstract what PhysDB-FCB is used for.

We have now added the following text in the abstract on the purpose of the PhysDB-FCP - “The purpose of the PhysDB-FCP is to provide a user friendly digital online structured tool that standardizes initial and follow up clinical assessments generating data that can be used for clinical decision making and support future research in physiotherapy for patients with chronic disease.”

The statements from 35-38 were difficult to read. Does this mean that PTs responded to questions on attitude towards 1) experiences with the PhysDB-FCP, 2) perceptions of patient experiences, 3) resources and support and 4) ideal frequency of use- using Likert scale, and yes or no. They also had a comment section to suggest areas requiring improvement on the scale. Can the authors revise?

We have revised this part of the abstract so that it is hopefully stands clearer. Please let us know if it requires further modifications – “Fifty of the 103 physiotherapists introduced to the PhysDB-FCP completed an anonymous online survey. Physiotherapists were asked Likert/categorical and yes/no questions on experiences with the PhysDB-FCP within their practice, perceptions of patient experiences, suitability of the resources and support provided by the PhysDB-FCP working group and the ideal administration frequency of the assessments within the PhysDB-FCP. Open ended feedback on possible improvements to the PhysDB-FCP was also collected.”

Authors should provide similar databases across the globe that could be compared with PhysBD-FCB? If there is none, the authors should state it.

We have provided references to a similar database in Norway and other related databases around the world. We would be happy to include more references if required. The new text reads “A similar database has recently been developed/tested in Norway, where patients receiving primary healthcare have physiotherapy management and are also assessed using patient-reported questionnaires on disability, pain and psychosocial factors [11]. Other databases have also been developed but often include different patient populations, that aren’t in primary care and do not include data from patient reported questionnaires combined with extensive physical assessments [8–10,12].” 

Even though the PhysDB-FCB description has been published in detail elsewhere, a summary description of the tool/database will be great. For instance, examples of the obligatory functional test (e,g outcome measures tools, SPPB etc.).

We have now added extra information about what was measured - “The PhysDB-FCP consisted of a physiotherapy assessment including demographic information (name, age, sex, assessment date, year of diagnosis), health status (pain (location and numeric rating), medication, height, mass, involuntary weight loss and speaking/swallowing problems), questions on daily functioning (information about personal and instrumental activities of daily living, use of and type of assistive device, dizziness and fatigue), obligatory functional tests (Timed up and go test, Sit to stand test, walking test (either 6 minute walk test, 10 meter walk test or 40 meter walk test) and box an block test (if indicated)) and documentation of a treatment plan (goal setting, type of care received (individual, team-based, combined, supervised/unsupervised), expected time(s)/effect(s) of treatment, patient approval of treatment plan and reporting of plan to General Practitioner) [13]. A patient survey, usually administered prior to the physiotherapy session, consisted of individual questions (on civil status, education, work status, amount of sick leave, fear of falling and sleep quality) and validated questionnaires (EQ- 5D- 5L, WHODAS 2.0—12 item, WHO-5) [13].”

What is the global implication of the findings in this study. The authors stated this, but they did a similar tool/database to compare?

We were uncertain if the reviewer seeks a more general formulation with comparison to other databases. However, we have tried to elaborate on this in the future directions section. Please let us know if we have misunderstood - “The importance of focusing on implementation setting and considering different interests of different stakeholders, meaningfulness of the content of an intervention/tool and the process of implementation (planning evaluation and reflection) is important and has been described in literature [18]. The current study and previous studies [11,13] show the challenges of implementing an assessment tool in primary care in a heterogeneous patient cohort. Similarly to the current study, in a previous study on a different database [11], therapists did not always use the proposed tool with only ≈50% of eligible patients being assessed with the tool. In that study [11], similar to our previous work [13], reasons for non-compliance were proposed but not formally investigated. The current study explains why implementing a database is challenging and provides reasons and potential issues that can be considered when implementing similar tools in the future.”

The figures were difficult to read.

We were uncertain if this comment referred to the actual figure itself or difficulty understanding the labels within the figure. We have assessed the figure and have changed the labels within the figures to try and make them more understandable and standalone. Please let us know if this needs to be further modified. 

Reviewer #2: If Physiotherapists are not very confident in the use of this database, there is a huge possibility of erroneous data input, coding, and misinterpretation of outputs. This paper could be strengthened if authors discuss the implication of their findings on the results of past, current research studies based on this database.

Thank you for the comment - We have added a paragraph to direct readers to these potential issues – “Not all physiotherapists were positive about using the PhysDB-FCP, and through our survey, we may have only captured those most engaged with the tool. Given this, there is the possibility that less motivated/engaged physiotherapists may have imputed/coded data erroneously. This would affect data quality within the database and will need to be acknowledged/considered in future studies using the database. With any database, there are possible issues with data quality, and this will contribute to extraneous data noise. However, in our study, participation was entirely voluntary, and if physiotherapists didn’t want to use the tool, they weren’t required to. As such, it is likely that less motivated/engaged clinicians, would have not filled in specific items or decided not to use the tool. Further, data for each patient appeared internally consistent (i.e. patients with slow timed up and go times had fewer sit to stand repetitions in 30 seconds) increasing our confidence in the data. Although the current study identified some issues, we are glad we have identified these. Having this knowledge is valuable in identifying potential sources for change that will likely increase future compliance and data quality. Changes such as a) integrating the PhysDB-FCP into current systems so that only one data input is required, b) having a shorter more basic tool allowing physiotherapists to complete the tool in 45 min (the time physiotherapists are compensated for), and c) increasing resources for education and ongoing support, will inevitably improve compliance, subsequent data quality and reduce missing data.”

---

## [Decision Letter · Decision Letter 1]

27 Sep 2021

PONE-D-21-09985R1Therapist perceptions of the Danish Physiotherapy Research Database for assessing patients with chronic diseasesPLOS ONE

Dear Dr. Næss-Schmidt,

Thank you for submitting your manuscript to PLOS ONE. After careful consideration, we feel that it has merit but does not fully meet PLOS ONE’s publication criteria as it currently stands. Therefore, we invite you to submit a revised version of the manuscript that addresses the points raised during the review process. Please submit your revised manuscript by Nov 11 2021 11:59PM. If you will need more time than this to complete your revisions, please reply to this message or contact the journal office at plosone@plos.org. Please include the following items when submitting your revised manuscript:A rebuttal letter that responds to each point raised by the academic editor and reviewer(s). You should upload this letter as a separate file labeled 'Response to Reviewers'.A marked-up copy of your manuscript that highlights changes made to the original version. You should upload this as a separate file labeled 'Revised Manuscript with Track Changes'.An unmarked version of your revised paper without tracked changes. You should upload this as a separate file labeled 'Manuscript'.If applicable, we recommend that you deposit your laboratory protocols in protocols.io to enhance the reproducibility of your results. Protocols.io assigns your protocol its own identifier (DOI) so that it can be cited independently in the future. For instructions see: https://journals.plos.org/plosone/s/submission-guidelines#loc-laboratory-protocols. Additionally, PLOS ONE offers an option for publishing peer-reviewed Lab Protocol articles, which describe protocols hosted on protocols.io. Read more information on sharing protocols at https://plos.org/protocols?utm_medium=editorial-email&utm_source=authorletters&utm_campaign=protocols.

We look forward to receiving your revised manuscript.

Kind regards,

Sinan Kardeş, M.D.

Academic Editor

PLOS ONE

Journal Requirements:

Additional Editor Comments (if provided):

Reviewers' comments:

Reviewer's Responses to Questions

**Comments to the Author**

1. If the authors have adequately addressed your comments raised in a previous round of review and you feel that this manuscript is now acceptable for publication, you may indicate that here to bypass the “Comments to the Author” section, enter your conflict of interest statement in the “Confidential to Editor” section, and submit your "Accept" recommendation.

Reviewer #1: All comments have been addressed

Reviewer #2: All comments have been addressed

2. Is the manuscript technically sound, and do the data support the conclusions?

Reviewer #1: Partly

Reviewer #2: Yes

3. Has the statistical analysis been performed appropriately and rigorously? 

Reviewer #1: Yes

Reviewer #2: Yes

4. Have the authors made all data underlying the findings in their manuscript fully available?

Reviewer #1: Yes

Reviewer #2: Yes

5. Is the manuscript presented in an intelligible fashion and written in standard English?

Reviewer #1: Yes

Reviewer #2: Yes

6. Review Comments to the Author

Reviewer #1: Thanks for attempting to address my comments. Just one correction: (a) line 79, can the authors change aren't to are not, (b) instead of using tables, can the authors describe Table 2 and Table 3 (because they can easily be described). I made this suggestion because the Tables are scanty so are the sections that the authors refer the reader to.

Reviewer #2: Thanks to authors for a detailed paragraph addressing initial concerns. I have no further comments on this paper.

7. PLOS authors have the option to publish the peer review history of their article (what does this mean?). If published, this will include your full peer review and any attached files.

Reviewer #1: No

Reviewer #2: No

---

## [Author Response · Author response to Decision Letter 1]

30 Sep 2021

Reply to reviewers

Based on the reviewer comments 2, we have made amendments to the manuscript. 

In the tracked changes document, yellow denotes insertions from the original submission and turquoise with a strike through indicates deletions from the original submission. 

Reviewer 1:

Thanks for attempting to address my comments. 

Thank you for the time taken to make the comments – We understand that peer review is voluntary and we appreciate the time it must have taken you to read, comment and (subsequently) improve our manuscript. 

Line 79, can the authors change aren't to are not

We have changed this

Instead of using tables, can the authors describe Table 2 and Table 3 (because they can easily be described). I made this suggestion because the Tables are scanty so are the sections that the authors refer the reader to.

Thank you for this suggestion. As suggested, we have placed the information from the tables in the text. 

Reviewer 2:

Thanks to authors for a detailed paragraph addressing initial concerns. I have no further comments on this paper.

Thank you

---

## [Decision Letter · Decision Letter 2]

19 Oct 2021

Therapist perceptions of the Danish Physiotherapy Research Database for assessing patients with chronic disease

PONE-D-21-09985R2

Dear Dr. Næss-Schmidt,

We’re pleased to inform you that your manuscript has been judged scientifically suitable for publication and will be formally accepted for publication once it meets all outstanding technical requirements.

Kind regards,

Sinan Kardeş, M.D.

Academic Editor

PLOS ONE

Additional Editor Comments (optional):

Reviewers' comments:

Reviewer's Responses to Questions

**Comments to the Author**

1. If the authors have adequately addressed your comments raised in a previous round of review and you feel that this manuscript is now acceptable for publication, you may indicate that here to bypass the “Comments to the Author” section, enter your conflict of interest statement in the “Confidential to Editor” section, and submit your "Accept" recommendation.

Reviewer #1: All comments have been addressed

Reviewer #2: All comments have been addressed

2. Is the manuscript technically sound, and do the data support the conclusions?

Reviewer #1: Yes

Reviewer #2: Yes

3. Has the statistical analysis been performed appropriately and rigorously? 

Reviewer #1: Yes

Reviewer #2: Yes

4. Have the authors made all data underlying the findings in their manuscript fully available?

Reviewer #1: No

Reviewer #2: Yes

5. Is the manuscript presented in an intelligible fashion and written in standard English?

Reviewer #1: Yes

Reviewer #2: Yes

6. Review Comments to the Author

Reviewer #1: Thanks for revising this manuscript. Can you describe Table 1, in the section above as well. Aside that congrats for the great work.

Reviewer #2: Author's response are satisfactory

This manuscript is acceptable in the current version. I have no further comments

7. PLOS authors have the option to publish the peer review history of their article (what does this mean?). If published, this will include your full peer review and any attached files.

Reviewer #1: No

Reviewer #2: No

---

## [Editor Report · Acceptance letter]

26 Oct 2021

PONE-D-21-09985R2 

Therapist perceptions of the Danish Physiotherapy Research Database for assessing patients with chronic disease 

Dear Dr. Næss-Schmidt:

I'm pleased to inform you that your manuscript has been deemed suitable for publication in PLOS ONE. Congratulations! Your manuscript is now with our production department. 

Kind regards, 

on behalf of

Dr. Sinan Kardeş 

Academic Editor

PLOS ONE